# Multi-Task Learning for Compositional Data via Sparse Network Lasso

**DOI:** 10.3390/e24121839

**Published:** 2022-12-17

**Authors:** Akira Okazaki, Shuichi Kawano

**Affiliations:** 1Graduate School of Informatics and Engineering, The University of Electro-Communications, 1-5-1 Chofugaoka, Chofu 182-8585, Tokyo, Japan; 2Faculty of Mathematics, Kyushu University, 744 Motooka, Nishi-ku 819-0395, Fukuoka, Japan

**Keywords:** clustering, log-contrast model, multi-task learning, symmetric form, variable selection

## Abstract

Multi-task learning is a statistical methodology that aims to improve the generalization performances of estimation and prediction tasks by sharing common information among multiple tasks. On the other hand, compositional data consist of proportions as components summing to one. Because components of compositional data depend on each other, existing methods for multi-task learning cannot be directly applied to them. In the framework of multi-task learning, a network lasso regularization enables us to consider each sample as a single task and construct different models for each one. In this paper, we propose a multi-task learning method for compositional data using a sparse network lasso. We focus on a symmetric form of the log-contrast model, which is a regression model with compositional covariates. Our proposed method enables us to extract latent clusters and relevant variables for compositional data by considering relationships among samples. The effectiveness of the proposed method is evaluated through simulation studies and application to gut microbiome data. Both results show that the prediction accuracy of our proposed method is better than existing methods when information about relationships among samples is appropriately obtained.

## 1. Introduction

Multi-task learning is a statistical methodology that assumes a different model for each task and jointly estimates these models. By sharing the common information between them, the generalization performance of estimation and prediction tasks is improved [1]. Multi-task learning has been used in various fields of research, such as computer vision [2], natural language processing [3], and life sciences [4]. In life sciences, the risk factors may vary from patient to patient [5], and a model that is common to all patients cannot sufficiently extract general risk factors. In multi-task learning, each patient can be considered as a single task, and different models are built for each patient to extract both patient-specific and common factors for the disease [6]. Localized lasso [7] is a method that performs multi-task learning using network lasso regularization [8]. By treating each sample as a single task, localized lasso simultaneously performs multi-task learning and clustering in the framework of a regression model.

On the other hand, compositional data, which consist of the individual proportions of a composition, are used in the fields of geology and life sciences for microbiome analysis. Compositional data are constrained to always take positive values summing to one. Due to these constraints, it is difficult to apply existing multi-task learning methods to compositional data. In the field of microbiome analysis, studies on gut microbiomes [9,10] have suggested that there are multiple types of gut microbiome clusters that vary from individual to individual [11]. In the case of such data where multiple clusters may exist, it is difficult to extract sufficient information using existing regression models for compositional data.

In this paper, we propose a multi-task learning method for compositional data, focusing on the network lasso regularization and the symmetric form of the log-contrast model [12], which is a linear regression model with compositional covariates. The symmetric form is extended to the locally symmetric form in which each sample has a different regression coefficient vector. These regression coefficient vectors are clustered by the network lasso regularization. Furthermore, because the dimensionality of features in compositional data has been increasing, in particular in microbiome analysis [13], we use an $\ell_{1}$-regularization [14] to perform variable selection. The advantage of using $\ell_{1}$-regularization is being able to perform variable selection even if the number of parameters exceeds the sample size. In addition, $\ell_{1}$-regularization is formulated by convex optimization, which leads to feasible computation, while classical subset selection is not. The estimation of the parameters included in the model is performed using an estimation algorithm based on the alternating direction method of multipliers [15], because the model includes non-differentiable points in the $\ell_{1}$-regularization term and zero-sum constraints on the parameters. The constructed model includes regularization parameters, which are determined by cross-validation (CV).

The remainder of this paper is organized as follows. Section 2 introduces multi-task learning based on a network lasso. In Section 3, we describe the regression models for compositional data. We propose a multi-task learning method for compositional data and its estimation algorithm in Section 4. In Section 5, we discuss the effectiveness of the proposed method through Monte Carlo simulations. An application to gut microbiome data is presented in Section 6. Finally, Section 7 summarizes this paper and discusses future work.

## 2. Multi-Task Learning Based on a Network Lasso

Suppose that we have *n* observed *p*-dimensional data $\left\{x_{i};i=1,\ldots ,n\right\}$ and *n* observed data for the response variable $\left\{y_{i};i=1,\ldots ,n\right\}$ and that these pairs $\left\{\left(y_{i},x_{i}\right),i=1,\ldots ,n\right\}$ are given independently. The graph $R=R^{T}\in R^{n\times n}$ is also given, where ${\left(R\right)}_{ij}=r_{i,j}\geq 0$ represents the relationship between the sample pair $\left(y_{i},x_{i}\right)$ and $\left(y_{j},x_{j}\right)$, and thus the diagonal components are zero.

We consider the following linear regression model:(1)$$y_{i}=x_{i}^{T}w_{i}+\epsilon_{i},\;i=1,\ldots ,n,$$
where $w_{i}={\left(w_{i1},\ldots ,w_{ip}\right)}^{T}\in R^{p}$ is the *p*-dimensional regression coefficient vector for sample $x_{i}$, and $\epsilon_{i}$ is an error term distributed as $N(0,\sigma^{2})$ independently. Note that we exclude the intercept $w_{0}$ from the model, because we assume the centered response and the standardized explanatory variables. Model (Equation 1) comprises a different model for each sample. In classical regression models, the regression coefficient vectors are assumed to be identical (i.e., $w_{1}=w_{2}=\cdots =w_{n}$).

For Model (Equation 1), we consider the following minimization problem:(2)$$\min_{w_{i}\in R^{p},\;i=1\ldots ,n}\left\{\sum_{i=1}^{n}{\left(y_{i}-x_{i}^{T}w_{i}\right)}^{2}+\lambda \sum_{m>l}^{n}r_{m,l}{∥w_{m}-w_{l}∥}_{2}\right\},$$
where λ(>0) is a regularization parameter. The second term in (Equation 2) is the network lasso regularization term [8]. For coefficient vectors $w_{m}$ and $w_{l}$, the network lasso regularization term induces $w_{m}=w_{l}$. If these vectors are estimated to be the same, then the *m*-th and *l*-th samples are interpreted as belonging to the same cluster. In the framework of multi-task learning, the minimization problem (Equation 2) considers one sample as one task by setting a coefficient vector for each sample. This allows us to extract the information of the regression coefficient vectors separately for each task. In addition, by clustering the regression coefficient vectors using the network lasso regularization term, we can extract the common information among tasks.

Yamada et al. [7] proposed the localized lasso for minimization problem (Equation 2) by adding an $\ell_{1,2}$-norm regularization term [16] as follows:(3)$$\min_{w_{i}\in R^{p},\;i=1,\ldots ,n}\left\{\sum_{i=1}^{n}{\left(y_{i}-x_{i}^{T}w_{i}\right)}^{2}+\lambda_{1}\sum_{m>l}^{n}r_{m,l}∥w_{m}-w_{l}{∥}_{2}+\lambda_{2}\sum_{i=1}^{n}{∥w_{i}∥}_{1}^{2}\right\}.$$ The $\ell_{1,2}$-norm regularization term induces group structure and intra-group level sparsity: several regression coefficients in a group are estimated to be zero, but at least one is estimated to be non-zero by squaring over the $\ell_{1}$-norm. In the localized lasso, each regression coefficient vector $w_{i}$ is treated as a group in order to remain $w_{i}\neq 0$. The localized lasso is used for multi-task learning and variable selection.

## 3. Regression Modeling for Compositional Data

The *p*-dimensional compositional data $x={\left(x_{1},\ldots ,x_{p}\right)}^{T}$ are defined as proportional data in the simplex space:(4)$$S^{p-1}=\left\{\left(x_{1},\ldots ,x_{p}\right):x_{j}>0\;\left(j=1,\ldots ,p\right),\sum_{j=1}^{p}x_{j}=1\right\}.$$ This structure imposes dependence between the features of the compositional data. Thus, statistical methods defined for spaces of real numbers cannot be applied [17]. To overcome this problem, Aitchison and Bacon-Shone [12] proposed the log-contrast model, which is a linear regression model with compositional covariates.

Suppose that we have *n* observed *p*-dimensional compositional data $\left\{x_{i};i=1,\ldots ,n\right\}$ and *n* objective variable data $\left\{y_{i};i=1,\ldots ,n\right\}$ and these pairs $\left\{\left(y_{i},x_{i}\right),i=1\ldots ,n\right\}$ are given independently. The log-contrast model is represented as follows:(5)$$y_{i}=\sum_{j=1}^{p-1}\log\frac{x_{ij}}{x_{ip}}\beta_{j}+\epsilon_{i},\;i=1,\ldots ,n,$$
where $\beta ={\left(\beta_{1},\ldots ,\beta_{p-1}\right)}^{T}\in R^{p-1}$ is a regression coefficient vector. Because the model uses an arbitrary variable as a reference for all other variables, the solution changes depending on the selection of the reference. By introducing $\beta_{p}=-\sum_{j=1}^{p-1}\beta_{j}$, the log-contrast model is equivalently expressed in symmetric form as:(6)$$y_{i}=z_{i}^{T}\beta +\epsilon_{i},\;s.t.\;\sum_{j=1}^{p}\beta_{j}=0,\;i=1,\ldots ,n,$$
where $z_{i}={\left(\log x_{i1},\ldots ,\log x_{ip}\right)}^{T}$, and $\beta ={\left(\beta_{1},\ldots ,\beta_{p}\right)}^{T}\in R^{p}$ is a regression coefficient vector. Lin et al. [13] proposed the minimization problem to select relevant variables in symmetric form by adding an $\ell_{1}$-regularization term [14]:(7)$$\min_{\beta \in R^{p}}\left\{\sum_{i=1}^{n}{\left(y_{i}-z_{i}^{T}\beta \right)}^{2}+\lambda {∥\beta ∥}_{1}\right\},\;s.t.\;\sum_{j=1}^{p}\beta_{j}=0.$$ Other models that extend this symmetric form of the problem have also been proposed [18,19,20,21].

## 4. Proposed Method

In this section, we propose a multi-task learning method for compositional data based on the network lasso and the symmetric form of the log-contrast model.

### 4.1. Model

We consider the locally symmetric form of the log-contrast model:(8)$$y_{i}=z_{i}^{T}w_{i}+\epsilon_{i},\;s.t.\;\sum_{j=1}^{p}w_{ij}=0,\;\;i=1,\ldots ,n,$$
where $z_{i}={\left(\log x_{i1},\ldots ,\log x_{ip}\right)}^{T}$, and $w_{i}={\left(w_{i1},\ldots ,w_{ip}\right)}^{T}$ is the regression coefficient vector for *i*-th sample of compositional data $x_{i}$. For Model (Equation 8), we consider the following minimization problem:(9)$$\begin{array}{c} \min_{w_{i}\in R^{p},\;i=1\ldots ,n}\left\{\sum_{i=1}^{n}{\left(y_{i}-z_{i}^{T}w_{i}\right)}^{2}+\lambda_{1}\sum_{m>l}^{n}r_{m,l}∥w_{m}-w_{l}{∥}_{2}+\lambda_{2}\sum_{i=1}^{n}{∥w_{i}∥}_{1}\right\},\\ s.t.\;\sum_{j=1}^{p}w_{ij}=0,\;i=1,\ldots ,n, \end{array}$$
where $\lambda_{1},\lambda_{2}\;\left(>0\right)$ are regularization parameters. The second term is the network lasso regularization term, which performs the clustering of the regression coefficient vectors. The third term is the $\ell_{1}$-regularization term [14]. This term is interpreted as a special case of the $\ell_{1,2}$-regularization term used in Model (Equation 3). Unlike the $\ell_{1}$-regularization term, it is difficult to optimize the $\ell_{1,2}$-regularization directly, because it does not have a closed form of the updates. To construct the estimation algorithm that performs variable selection and preserves the constraints for regression coefficient vectors simultaneously, we employ the $\ell_{1}$-regularization term. Since variable selection is performed by the $\ell_{1}$-regularization term, we refer to the combination of the second term and the third term as sparse network lasso after sparse group lasso [22].

For Model (8), when a new data point $z_{i^{*}}$ is obtained after the estimation, there is no corresponding regression coefficient vector $w_{i^{*}}$ for $z_{i^{*}}$. Thus, it is necessary to estimate the coefficient vector for predicting the response. Hallac et al. [8] proposed solving the following minimization problem:(10)$$\min_{w_{i^{*}}\in R^{p}}\left\{\sum_{i=1}^{n}r_{i^{*},i}{∥w_{i^{*}}-{\hat{w}}_{i}∥}_{2}\right\},$$
where ${\hat{w}}_{i}$ is the estimated regression coefficient vector for the *i*-th sample. This problem is also known as the Weber problem. The solution of this problem is interpreted as the weighted geometric median of ${\hat{w}}_{i}$. For our proposed method, we consider solving the constrained Weber problem with the zero-sum constraint in the form:(11)$$\min_{w_{i^{*}}\in R^{p}}\left\{\sum_{i=1}^{n}r_{i^{*},i}{∥w_{i^{*}}-{\hat{w}}_{i}∥}_{2}\right\},\;s.t.\sum_{j=1}^{p}w_{i^{*}j}=0.$$

### 4.2. Estimation Algorithm

To construct the estimation algorithm for the proposed method, we rewrite minimization problem (Equation 9) as follows:(12)$$\begin{array}{c} \min_{w_{i},a_{m},b_{i}\in R^{p},\;i=1\ldots ,n}\left\{\sum_{i=1}^{n}{\left(y_{i}-z_{i}^{T}w_{i}\right)}^{2}+\lambda_{1}\sum_{m>l}^{n}r_{m,l}∥a_{m,l}-a_{l,m}{∥}_{2}+\lambda_{2}\sum_{i=1}^{n}{∥b_{i}∥}_{1}\right\},\\ s.t.\;w_{m}=a_{m,l},\;w_{i}=b_{i},\;1_{p}^{T}w_{i}=0,\;i,m,l=1,\ldots ,n, \end{array}$$
where $1_{p}$ is the *p*-dimensional vector of ones. The augmented Lagrangian for (Equation 12) is formulated as:(13)$$\begin{array}{cc} L{\left(\Theta ,Q\right)}_{\Omega } & =\sum_{i=1}^{n}{\left(y_{i}-z_{i}^{T}w_{i}\right)}^{2}\\ \; & +\sum_{m>l}^{n}\left\{\lambda_{1}r_{m,l}{∥a_{m,l}-a_{l,m}∥}_{2}\right.\\ \; & +\left.\frac{\rho }{2}(∥w_{m}-a_{m,l}+s_{m,l}{∥}_{2}^{2}+∥w_{l}-a_{l,m}+s_{l,m}{∥}_{2}^{2})-\frac{\rho }{2}(∥s_{m,l}{∥}_{2}^{2}+∥s_{l,m}{∥}_{2}^{2})\right\}\\ \; & +\sum_{i=1}^{n}\left\{\lambda_{2}∥b_{i}{∥}_{1}+t_{i}^{T}\left(w_{i}-b_{i}\right)+\frac{\phi }{2}{∥w_{i}-b_{i}∥}_{2}^{2}\right\}\\ \; & +\sum_{i=1}^{n}\left\{u_{i}1_{p}^{T}w_{i}+\frac{\psi }{2}{∥1_{p}^{T}w_{i}∥}_{2}^{2}\right\}, \end{array}$$
where $s_{m,l},t_{i},u_{i}$ are the Lagrange multipliers and ρ,ϕ,ψ(>0) are the tuning parameters. For simplicity of notation, the parameters in the model $w_{i},a_{i,j},b_{i}$ are collectively denoted as Θ, the Lagrange multipliers are collectively denoted as *Q*, and the tuning parameters are collectively denoted as Ω.

The update algorithm for Θ with the alternating direction method of multipliers (ADMM) is obtained from the following minimization problem:(14)$$\begin{array}{cc} w^{\left(k+1\right)} & =\underset{w}{\arg\;\min}L{\left(w,a^{\left(k\right)},b^{\left(k\right)},Q^{\left(k\right)}\right)}_{\Omega },\\ a^{\left(k+1\right)} & =\underset{a}{\arg\;\min}L{\left(w^{\left(k+1\right)},a,b^{\left(k\right)},Q^{\left(k\right)}\right)}_{\Omega },\\ b^{\left(k+1\right)} & =\underset{b}{\arg\;\min}L{\left(w^{\left(k+1\right)},a^{\left(k+1\right)},b,Q^{\left(k\right)}\right)}_{\Omega }, \end{array}$$
where *k* denotes the repetition number. By repeating the updates (Equation 14) and the update for *Q*, the estimation algorithm for (Equation 12) is represented by Algorithm 1. The estimation algorithm for (Equation 11) is represented by Algorithm 2 with the update from ADMM. The details of the derivations of the estimation algorithms are presented in Appendices Appendix A and Appendix B.
**Algorithm 1** Estimation algorithm for (Equation 12) via ADMM**Require:** Initialize $w^{\left(0\right)},a^{\left(0\right)},b^{\left(0\right)},s^{\left(0\right)},t^{\left(0\right)},u^{\left(0\right)}$. **while** convergence **do**    **for**  i=1…,n **do**            
$$\begin{array}{cc} w_{i}^{\left(k+1\right)} & ={\left\{2z_{i}z_{i}^{T}+\left(\rho \left(n-1\right)+\phi \right)I_{p}+\psi 1_{p}1_{p}^{T}\right\}}^{-1}\\ \; & \left\{2y_{i}z_{i}+\rho \sum_{m\neq i}^{n}\left(a_{i,m}^{\left(k\right)}-s_{i,m}^{\left(k\right)}\right)-t_{i}^{\left(k\right)}+\phi b_{i}^{\left(k\right)}-u_{i}^{\left(k\right)}1_{p}\right\} \end{array}$$     **end for**     **for** m,l=1,…,n,(m>l) **do**        $\theta =\max\left(1-\frac{\lambda_{1}r_{m,l}}{\rho ∥\left(w_{m}^{\left(k+1\right)}+s_{m,l}^{\left(k\right)}\right)-\left(w_{l}^{\left(k+1\right)}+s_{l,m}^{\left(k\right)}\right){∥}_{2}},0.5\right)$        $a_{m,l}^{\left(k+1\right)}=\theta \left(w_{m}^{\left(k+1\right)}+s_{m,l}^{\left(k\right)}\right)+\left(1-\theta \right)\left(w_{l}^{\left(k+1\right)}+s_{l,m}^{\left(k\right)}\right)$        $a_{l,m}^{\left(k+1\right)}=\left(1-\theta \right)\left(w_{m}^{\left(k+1\right)}+s_{m,l}^{\left(k\right)}\right)+\theta \left(w_{l}^{\left(k+1\right)}+s_{l,m}^{\left(k\right)}\right)$    **end for**    **for** i=1,…,n,j=1,…,p **do**        $b_{ij}^{\left(k+1\right)}=S\left(w_{ij}^{\left(k+1\right)}+\frac{1}{\phi }t_{ij}^{\left(k\right)},\frac{\lambda_{2}}{\phi }\right)$    **end for**    **for** m,l=1,…,n,(m≠l) **do**        $s_{m,l}^{\left(k+1\right)}=s_{m,l}^{\left(k\right)}+\rho \left(w_{m}^{\left(k+1\right)}-a_{m,l}^{\left(k+1\right)}\right)$    **end for**    **for** i=1…,n **do**        $t_{i}^{\left(k+1\right)}=t_{i}^{\left(k\right)}+\phi \left(w_{i}^{\left(k+1\right)}-b_{i}^{\left(k+1\right)}\right)$        $u_{i}^{\left(k+1\right)}=u_{i}^{\left(k\right)}+\psi 1_{p}^{T}w_{i}^{\left(k+1\right)}$    **end for****end while****Ensure:** $b_{i},\;i=1,\ldots ,n.$

**Algorithm 2** Estimation algorithm for constrained Weber problem (Equation 11) via ADMM
**Require:** Initialize $w_{i^{*}}^{\left(0\right)},e^{\left(0\right)},u^{\left(0\right)},v^{\left(0\right)}$.**while** convergence **do**    **for** i=1…,n **do**        $e_{i}^{\left(k+1\right)}=\min\left(\frac{r_{i^{*},i}}{\mu },{∥w_{i^{*}}^{\left(k\right)}-\frac{1}{\mu }u_{i}^{\left(k\right)}-{\hat{w}}_{i}∥}_{2}\right)\frac{w_{i^{*}}^{\left(k\right)}-\frac{1}{\mu }u_{i}^{\left(k\right)}-{\hat{w}}_{i}}{∥w_{i^{*}}^{\left(k\right)}-\frac{1}{\mu }u_{i}^{\left(k\right)}-{\hat{w}}_{i}{∥}_{2}}$    **end for**    $w_{i^{*}}^{\left(k+1\right)}={\left(\mu nI_{p}+\eta 1_{p}1_{p}^{T}\right)}^{-1}\left\{\mu \sum_{i=1}^{n}\left(e_{i}^{\left(k+1\right)}+\frac{1}{\mu }u_{i}^{\left(k\right)}\right)-v^{\left(k\right)}1_{p}\right\}$    **for** i=1…,n **do**        $u_{i}^{\left(k+1\right)}=u_{i}^{\left(k\right)}+\mu \left(e_{i}^{\left(k+1\right)}-w_{i^{*}}^{\left(k+1\right)}\right)$    **end for**    $v^{\left(k+1\right)}=v^{\left(k\right)}+\eta 1_{p}^{T}{w_{i^{*}}}^{\left(k+1\right)}$
**end while**
**Ensure:**  $w_{i^{*}}$


## 5. Simulation Studies

In this section, we report simulation studies conducted with our proposed method using artificial data.

In our simulations, we generated artificial data from the true model:(15)$$y_{i}=\left\{\begin{array}{cc} z_{i}^{T}w_{\left(1\right)}^{*}+\epsilon_{i}, & (i=1,\ldots ,40),\\ z_{i}^{T}w_{\left(2\right)}^{*}+\epsilon_{i}, & (i=41,\ldots ,80),\\ z_{i}^{T}w_{\left(3\right)}^{*}+\epsilon_{i}, & (i=81,\ldots ,120), \end{array}\right.$$
where $z_{i}={\left(\log x_{i1},\ldots ,\log x_{ip}\right)}^{T}$, $x_{i}={\left(x_{i1},\ldots ,x_{ip}\right)}^{T}$ is *p*-dimensional compositional data, $w_{\left(1\right)}^{*}$,$w_{\left(2\right)}^{*}$,$w_{\left(3\right)}^{*}\in R^{p}$ are the true regression coefficient vectors, and $\epsilon_{i}$ is an error term distributed as $N(0,\sigma^{2})$ independently. We generated compositional data $\left\{x_{i};i=1,\ldots ,120\right\}$ as follows. First, we generated the data $\left\{c_{i},i=1,\ldots ,120\right\}$ from the *p*-dimensional multivariate normal distribution $N_{p}\left(\omega ,\Sigma \right)$ independently, where ${\left(\omega \right)}_{j}=\omega_{j}$, ${\left(\Sigma \right)}_{ij}=0.2^{\left|i-j\right|}$, and
(16)$$\omega_{j}=\left\{\begin{array}{cc} \log(0.5p), & (j=1,\ldots ,5),\\ 0, & (j=6,\ldots ,p). \end{array}\right.$$ Then, the data $\left\{c_{i},i=1,\ldots ,120\right\}$ were converted to the compositional data $\left\{x_{i};i=1,\ldots ,120\right\}$ by the following transformation:(17)$$x_{ij}=\frac{\exp(c_{ij})}{\sum_{k=1}^{p}\exp\left(c_{ik}\right)},\;i=1,\ldots ,120,\;j=1,\ldots ,p.$$ The true regression coefficient vectors were set as:(18)$$\begin{array}{cc} w_{\left(1\right)}^{*} & ={\left(1,-0.8,0.6,0,0,-1.5,-0.5,1.2,0_{p-8}^{T}\right)}^{T},\\ w_{\left(2\right)}^{*} & ={\left(0,-0.5,1,1.2,0.1,-1,0,-0.8,0_{p-8}^{T}\right)}^{T},\\ w_{\left(3\right)}^{*} & ={\left(0,0,0,0.8,1,0,-0.8,-1,0_{p-8}^{T}\right)}^{T}. \end{array}$$ We also assumed that the graph $R\in {\left\{0,1\right\}}^{120\times 120}$ was observed. In the graph, the true value of each element was obtained with probability $P_{R}$. We calculated MSE as $\sum_{i^{*}=1}^{n^{*}}{\left(y_{i^{*}}-z_{i^{*}}^{T}{\hat{w}}_{i^{*}}\right)}^{2}/n^{*}$, dividing the 120 samples into 100 training data and 20 validation data. Here, $n^{*}$ indicates the number of samples for validation data (i.e., 20). The regression coefficient vectors for the validation data were estimated based on the constrained Weber problem (Equation 11). To evaluate the effectiveness of our proposed method, it is compared with both Model (Equation 7) and the model obtained by removing the zero-sum constraint from minimization problem (Equation 9). We refer to the latter two comparison methods as compositional lasso (CL) and sparse network lasso (SNL), respectively. To the best of our knowledge, there are no studies that simultaneously perform regression and clustering on compositional data. Therefore, we compared with the CL and SNL models; CL assumes the situation where the existence of the multiple clusters is not considered, while SNL considers their existence while the nature of the compositional data is ignored.

The regularization parameters were determined by five-fold CV for both the proposed method and the comparison methods. The values of tuning parameters ρ,ϕ,ψ,μ,η for ADMM were all set to one. We considered several settings: p={30,100,200}, σ={0.1,0.5,1}, $P_{R}=\left\{0.99,0.95,0.90,0.80,0.70\right\}$. We generated 100 datasets and computed the mean and standard deviation of MSE from the 100 repetitions.

Table 1, Table 2 and Table 3 show the results for the mean and standard deviation of MSE for each σ. The proposed method and SNL show better prediction accuracy than CL in almost settings. The reason for this may be that CL assumes only a single regression coefficient vector and thus fails to capture the true structure, which consists of three clusters. A comparison between the proposed method and SNL shows that the proposed method has higher prediction accuracy than SNL when $P_{R}=0.99,0.95,$ and 0.90, whereas SNL shows better results in most cases when $P_{R}=0.80,0.70$. This means that the proposed method is superior to SNL when the structure of the graph *R* is similar to the true structure. On the whole, the prediction accuracy deteriorates as $P_{R}$ decreases for both the proposed method and SNL, but this deterioration is more extreme for the proposed method. For both the proposed method and SNL, which assume multiple regression coefficient vectors, the standard deviation is somewhat large.

## 6. Application to Gut Microbiome Data

The gut microbiome affects human health/disease, for example, in terms of obesity. Gut microbiomes may be exposed to inter-individual heterogeneity from environmental factors such as diet as well as from hormonal factors and age [23,24]. In this section, we applied our proposed method to the real dataset reported by [9]. This dataset was obtained from a cross-sectional study of 98 healthy volunteers conducted at the University of Pennsylvania to investigate the connections between long-term dietary patterns and gut microbiome composition. Stool samples were collected from the subjects, and DNA samples were analyzed by 454/Roche pyrosequencing of 16S rRNA gene segments of the V1–V2 region. In the results, the counts for more than 17,000 species-level OTUs were obtained. Demographic data, such as body mass index (BMI), sex, and age, were also obtained.

We used centered BMI as the response and the compositional data of the gut microbiome as the explanatory variable. In order to reduce their number, we used single-linkage clustering based on an available phylogenetic tree to aggregate the OTUs, which is provided as the function **tip_glom** in the R package “phyloseq” see [25]. In this process, some closely related OTUs defined on the leaf nodes of the phylogenetic tree are aggregated into one OTU when the cophenetic distances between the OTUs are smaller than a certain threshold. We set the threshold at 0.5. As a result, 166 OTUs were obtained after the aggregation. Since some OTUs had zero counts, making it impossible to take the logarithm, these counts were replaced by the value one before converting them to compositional data.

We computed the graph $R\in R^{98\times 98}$ as follows: (19)$$R=\frac{S^{T}+S}{2},\;{\left(S\right)}_{ij}=\left\{\begin{array}{cc} 1 & j\mathrm{-th}\mathrm{sample}\mathrm{is}a\;5\mathrm{-NN}\mathrm{of}\;i\mathrm{-th}\mathrm{sample}\mathrm{with}\;D_{ij},\\ 0 & \mathrm{Otherwise}, \end{array}\right.$$
where $D_{ij}$ is the distance between the *i*-th and *j*-th samples. Distance $D_{ij}$ was calculated in the following two ways: (i) Gower distance [26] was calculated using the sex and age data of the subjects. (ii) Log-ratio distance (e.g., see [27]) was calculated using the explanatory variable, as follows:(20)$$D_{ij}=\sqrt{\sum_{l=1}^{p}{\left(z_{il}^{c}-z_{jl}^{c}\right)}^{2}},$$
where $z_{ij}^{c}=\log x_{ij}-\frac{1}{p}\sum_{j=1}^{p}\log x_{ij}$. Using these two ways of calculating distance, we obtained two different *R* and estimation results. We refer to these two methods as Proposed (i) and Proposed (ii), respectively. Equation (Equation 19) is the same as the one used in Yamada et al. [7] in the application to real datasets.

To evaluate the prediction accuracy of our proposed method, we calculated MSE by randomly splitting the dataset into 90 samples as the training data and eight samples as the validation data. We again used the method of Lin et al. (2014), which is referred to as compositional lasso (CL), as a comparison method. The regularization parameters were determined by five-fold CV for both our proposed method and CL. The mean and standard deviation of MSE were calculated from 100 repetitions.

Table 4 shows the mean and standard deviation of MSE in the real data analysis. We observe that Proposed (i) has the smallest mean and standard deviation of MSE. This indicates that our proposed method captures the inherent structure of the compositional data by providing an appropriate graph *R*. However, the standard deviation is large even for Proposed (i), which indicates that the prediction accuracy may strongly depend on the assignments of samples to the training data and the validation data.

Table 5 shows the coefficient of determination $R^{2}$ using leave-one-out cross-validation (LOOCV) for Proposed (i) and CL. The fittings of the observed and predicted BMI are plotted in Figure 1a,b for Proposed (i) and CL, respectively. The horizontal axis represents the centered observed BMI values, and the vertical axis represents the corresponding predicted BMI. As shown, CL does not predict data with centered observed values between −10 and −5 as being in that interval, whereas Proposed (i) predicts these data correctly to some extent.

Figure 2 shows the regression coefficients estimated by Proposed (i) for all samples, where the regularization parameters are determined by LOOCV. To obtain the results in Figure 2, we used hierarchical clustering to group together similar regression coefficient vectors, setting the number of clusters as five.

With our proposed method, many of the estimated regression coefficients were not exactly zero but close to zero. Thus, we will treat estimated regression coefficients $|{\hat{w}}_{ij}|<0.05$ as being exactly zero to simplify the interpretation. Figure 3 shows only those coefficients that satisfy $|{\hat{w}}_{ij}|\geq 0.05$ in at least one sample, where the corresponding variables are listed in Table 6.

It is reported that the human gut microbiome can be classified into three clusters, called enterotypes, which are characterized by three predominant genera: *Bacteroides*, *Prevotella*, and *Ruminococcus* [11]. In the dataset, OTUs of genus levels *Prevotella* and *Ruminococcus* were aggregated into the OTUs of family levels *Prevotellaceae* and *Ruminococcaceae* by the single-linkage clustering. In Figure 3, *Bacteroides* correspond to OTU5, 6, 7, 8, 9, and 10; *Prevotellaceae* corresponds to OTU12; and *Ruminococcaceae* corresponds to OTU30 and 31. For these OTUs, the differences are clear between OTU6, 9, 10 and OTU30, 31 among samples 81–90, in which only *Bacteroides* are correlated to the response. On the other hand, the differences among samples 65–74 are also indicated, in which only *Bacteroides* do not affect BMI. These results suggest that *Bacteroides*, *Prevotellaceae*, and *Ruminococcaceae* may have different effects on BMI that are associated with enterotypes. In addition, it is reported that women with a higher abundance of *Prevotellaceae* are more obese [28]. The regression coefficients of non-zero *Prevotellaceae* are all positive, and the eight corresponding samples are all females. On the other hand, in OTU29 indicating *Roseburia*, 9 samples out of 10 are negatively associated with BMI. *Roseburia* is also reported to be negatively correlated with indicators of body weight [29].

## 7. Conclusions

We proposed a multi-task learning method for compositional data based on a sparse network lasso and log-contrast model. By imposing a zero-sum constraint on the model corresponding to each sample, we could extract the information of latent clusters in the regression coefficient vectors for compositional data. In the results of simulations, the proposed method worked well when clusters existed for the compositional data and an appropriate graph *R* was obtained. In a human gut microbiome analysis, our proposed method is better than the existing method in prediction accuracy by considering the heterogeneity from age and sex. In addition, cluster-specific OTUs such as ones related to enterotypes were detected in terms of effects on BMI.

Although our proposed method shrinks some regression coefficients that do not affect response to zero, many coefficients close to zero remain. Furthermore, in both the simulations and human gut microbiome analysis, the prediction accuracy of the proposed method deteriorated significantly when the obtained *R* did not capture the true structure. Moreover, the standard deviations of MSE were high in almost all cases. We leave these as topics of future research.

## Figures and Tables

**Figure 1 entropy-24-01839-f001:**
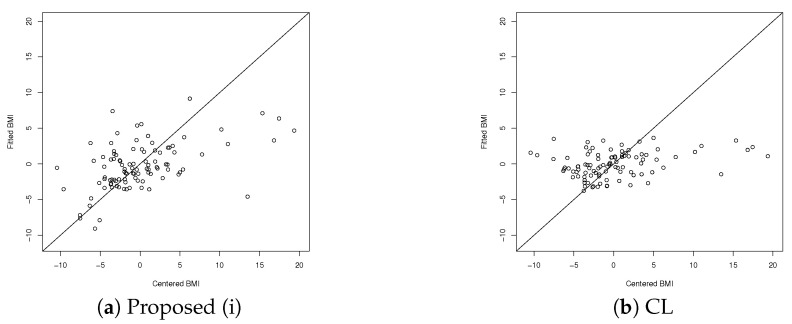
Observed and predicted BMI using LOOCV.

**Figure 2 entropy-24-01839-f002:**
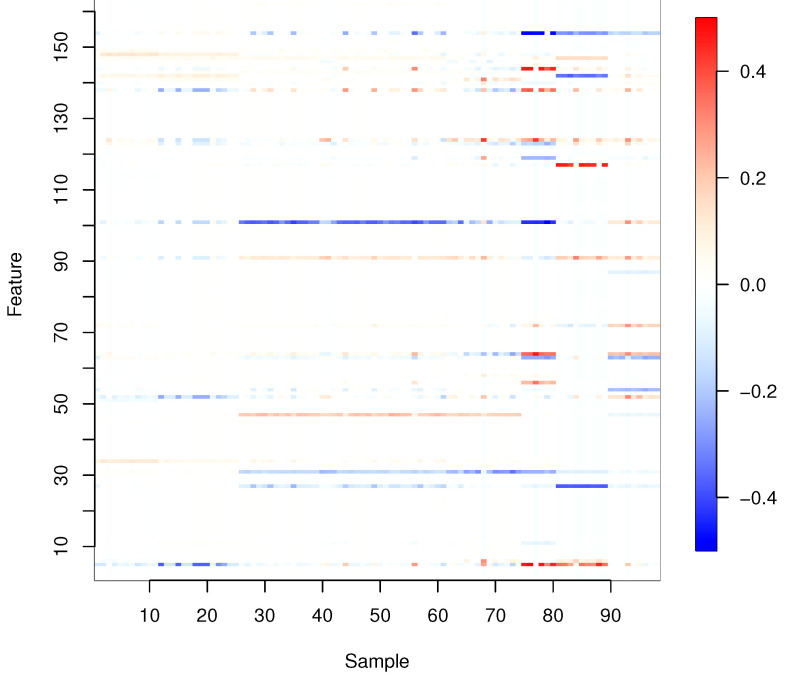
Estimated regression coefficients for all samples.

**Figure 3 entropy-24-01839-f003:**
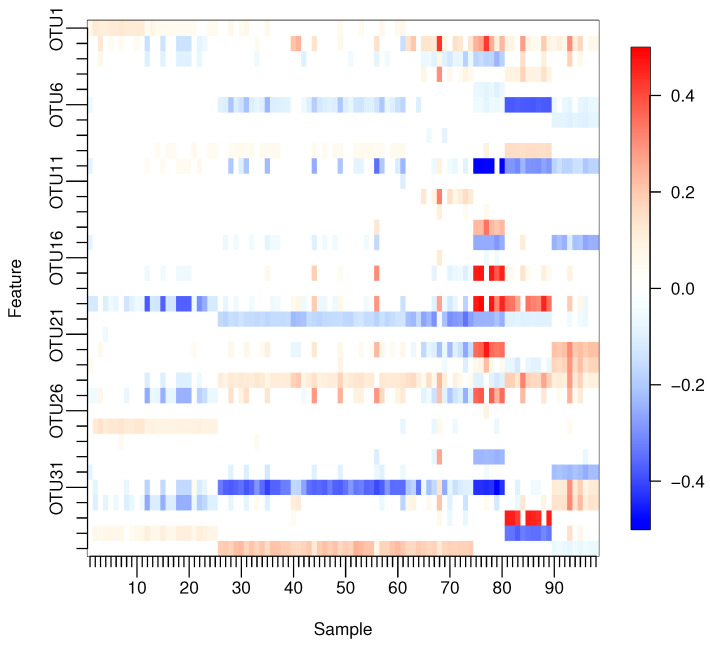
Only the estimated regression coefficients with $|{\hat{w}}_{ij}|\geq 0.05$ for at least one sample.

**Table 1 entropy-24-01839-t001:** Mean and deviation of MSE in σ=0.1 for simulations.

Method	p=30	p=100	p=200
CL	5.20(1.44)	6.99(1.81)	8.75(2.69)
$P_{R}=0.99$			
Proposed	0.51(0.58)	2.13(2.35)	2.70(1.92)
SNL	2.54(0.97)	3.73(1.31)	6.55(3.94)
$P_{R}=0.95$			
Proposed	2.74(1.19)	2.96(1.33)	3.68(2.71)
SNL	3.19(0.98)	3.87(1.35)	5.26(4.15)
$P_{R}=0.90$			
Proposed	3.29(1.38)	3.80(1.33)	4.49(1.81)
SNL	3.40(1.23)	4.25(1.49)	4.75(1.49)
$P_{R}=0.80$			
Proposed	4.20(1.58)	5.53(2.30)	7.49(3.98)
SNL	3.87(1.41)	4.86(1.52)	5.70(2.00)
$P_{R}=0.70$			
Proposed	4.13(1.56)	6.57(2.55)	7.66(2.28)
SNL	4.56(1.55)	5.63(1.72)	6.69(2.25)

**Table 2 entropy-24-01839-t002:** Mean and deviation of MSE in σ=0.5 for simulations.

Method	p=30	p=100	p=200
CL	5.64(1.42)	7.94(2.31)	10.02(2.96)
$P_{R}=0.99$			
Proposed	1.02(0.75)	2.13(1.50)	3.07(1.61)
SNL	2.97(1.23)	3.73(1.32)	5.98(3.95)
$P_{R}=0.95$			
Proposed	3.05(1.28)	3.36(1.05)	4.37(3.47)
SNL	3.50(1.20)	4.19(1.35)	5.18(2.54)
$P_{R}=0.90$			
Proposed	3.55(1.40)	4.83(1.61)	5.13(2.99)
SNL	3.83(1.30)	4.43(1.26)	5.21(3.42)
$P_{R}=0.80$			
Proposed	4.10(1.47)	5.21(1.89)	6.70(2.51)
SNL	4.06(1.35)	5.32(1.88)	6.06(1.94)
$P_{R}=0.70$			
Proposed	4.33(1.39)	6.59(2.62)	8.58(3.16)
SNL	4.53(1.54)	5.71(1.78)	7.37(2.24)

**Table 3 entropy-24-01839-t003:** Mean and deviation of MSE in σ=1 for simulations.

Method	p=30	p=100	p=200
CL	6.50(1.78)	8.05(2.57)	10.41(3.10)
$P_{R}=0.99$			
Proposed	2.34(1.29)	3.25(1.87)	3.87(1.97)
SNL	3.94(1.47)	4.98(1.67)	5.90(3.12)
$P_{R}=0.95$			
Proposed	3.43(1.24)	3.96(1.61)	4.73(2.35)
SNL	4.36(1.31)	4.73(1.29)	5.39(1.64)
$P_{R}=0.90$			
Proposed	4.28(1.55)	4.83(1.61)	5.09(1.88)
SNL	4.71(1.49)	5.25(2.08)	5.75(2.29)
$P_{R}=0.80$			
Proposed	5.67(1.80)	6.61(2.05)	7.77(3.68)
SNL	5.20(1.79)	6.06(2.01)	6.77(2.07)
$P_{R}=0.70$			
Proposed	5.73(1.87)	8.21(2.71)	8.86(3.57)
SNL	5.31(1.84)	7.02(2.29)	7.97(2.56)

**Table 4 entropy-24-01839-t004:** Mean and standard deviation of MSE for real data analysis (100 repetitions).

Value	Proposed (i)	Proposed (ii)	CL
MSE (SD)	23.01(16.62)	31.59(22.44)	30.96(23.36)

**Table 5 entropy-24-01839-t005:** Coefficients of determination using LOOCV.

Value	Proposed (i)	CL
LOOCV $R^{2}$	0.245	0.083

**Table 6 entropy-24-01839-t006:** Variables with estimated regression coefficients $|{\hat{w}}_{ij}|\geq 0.05$ for at least one sample.

Variable	Kingdom	Phylum	Class	Order	Family	Genus	Species
OTU1	*Bacteria*						
OTU2	*Bacteria*						
OTU3	*Bacteria*	*Bacteroidetes*					
OTU4	*Bacteria*	*Bacteroidetes*	*Bacteroidetes*	*Bacteroidales*			
OTU5	*Bacteria*	*Bacteroidetes*	*Bacteroidetes*	*Bacteroidales*	*Bacteroidaceae*	*Bacteroides*	
OTU6	*Bacteria*	*Bacteroidetes*	*Bacteroidetes*	*Bacteroidales*	*Bacteroidaceae*	*Bacteroides*	
OTU7	*Bacteria*	*Bacteroidetes*	*Bacteroidetes*	*Bacteroidales*	*Bacteroidaceae*	*Bacteroides*	
OTU8	*Bacteria*	*Bacteroidetes*	*Bacteroidetes*	*Bacteroidales*	*Bacteroidaceae*	*Bacteroides*	
OTU9	*Bacteria*	*Bacteroidetes*	*Bacteroidetes*	*Bacteroidales*	*Bacteroidaceae*	*Bacteroides*	
OTU10	*Bacteria*	*Bacteroidetes*	*Bacteroidetes*	*Bacteroidales*	*Bacteroidaceae*	*Bacteroides*	
OTU11	*Bacteria*	*Bacteroidetes*	*Bacteroidetes*	*Bacteroidales*	*Porphyromonadaceae*	*Parabacteroides*	
OTU12	*Bacteria*	*Bacteroidetes*	*Bacteroidetes*	*Bacteroidales*	*Prevotellaceae*		
OTU13	*Bacteria*	*Firmicutes*	*Clostridia*				
OTU14	*Bacteria*	*Firmicutes*	*Clostridia*	*Clostridiales*			
OTU15	*Bacteria*	*Firmicutes*	*Clostridia*	*Clostridiales*			
OTU16	*Bacteria*	*Firmicutes*	*Clostridia*	*Clostridiales*			
OTU17	*Bacteria*	*Firmicutes*	*Clostridia*	*Clostridiales*			
OTU18	*Bacteria*	*Firmicutes*	*Clostridia*	*Clostridiales*			
OTU19	*Bacteria*	*Firmicutes*	*Clostridia*	*Clostridiales*	*Lachnospiraceae*		
OTU20	*Bacteria*	*Firmicutes*	*Clostridia*	*Clostridiales*	*Lachnospiraceae*		
OTU21	*Bacteria*	*Firmicutes*	*Clostridia*	*Clostridiales*	*Lachnospiraceae*		
OTU22	*Bacteria*	*Firmicutes*	*Clostridia*	*Clostridiales*	*Lachnospiraceae*		
OTU23	*Bacteria*	*Firmicutes*	*Clostridia*	*Clostridiales*	*Lachnospiraceae*		
OTU24	*Bacteria*	*Firmicutes*	*Clostridia*	*Clostridiales*	*Lachnospiraceae*		
OTU25	*Bacteria*	*Firmicutes*	*Clostridia*	*Clostridiales*	*Lachnospiraceae*		
OTU26	*Bacteria*	*Firmicutes*	*Clostridia*	*Clostridiales*	*Lachnospiraceae*		
OTU27	*Bacteria*	*Firmicutes*	*Clostridia*	*Clostridiales*	*Lachnospiraceae*		
OTU28	*Bacteria*	*Firmicutes*	*Clostridia*	*Clostridiales*	*Lachnospiraceae*		
OTU29	*Bacteria*	*Firmicutes*	*Clostridia*	*Clostridiales*	*Lachnospiraceae*	*Roseburia*	
OTU30	*Bacteria*	*Firmicutes*	*Clostridia*	*Clostridiales*	*Ruminococcaceae*		
OTU31	*Bacteria*	*Firmicutes*	*Clostridia*	*Clostridiales*	*Ruminococcaceae*		
OTU32	*Bacteria*	*Firmicutes*	*Erysipelotrichia*	*Erysipelotrichales*	*Erysipelotrichaceae*	*Catenibacterium*	
OTU33	*Bacteria*	*Firmicutes*	*Erysipelotrichia*	*Erysipelotrichales*	*Erysipelotrichaceae*	*Erysipelotrichaceae.* *Incertae.Sedis*	
OTU34	*Bacteria*	*Proteobacteria*					
OTU35	*Bacteria*	*Proteobacteria*	*Gammaproteobacteria*	*Enterobacteriales*	*Enterobacteriaceae*		

## Data Availability

The data that support the findings of this study are available from the corresponding author, A.O., upon reasonable request. The source codes of the proposed method are available at https://github.com/aokazaki255/CSNL.

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
