# Peer review of "Multi-Task Learning for Compositional Data via Sparse Network Lasso"

_entropy, 2022, doi:10.3390/e24121839_

Round 1

Reviewer 1 Report

This manuscript introduced a multi-tas, learning method when explanatory variables include compositional data, using a sparse network lasso and focus on a symmetric form of the log-contrast model. By numerical experiments, they showed that the proposed method worked well in terms of prediction accuracy when an appropriate graph was obtained, comparing to two other method. Further, the proposed method was applied to analyze gut microbiome data. 

Since he proposed multi-task learning and regression method can deal with compositional data, and incorporates a sparse network lasso to facilitate variable selection, the proposed method seems useful and interesting.  Further, this paper provides a clear explanation of the of the methods used. Therefore, this manuscript may attract many readers of this journal. 

 Followings are some comments about minor points.

・line 10, "various fields of research" → you mentioned only life science. It seems bettter to include more references for other fields to say so. 

・line 11, "factors of a disease" → "risk factors" is more familiar.

・Z in equation (7) is the same as in (5)?  

・in equation (11), wTZ should be ZTw?

・line 99, what are CL and SNL?

・line 106 and 107, there are almost same sentences. you should pick one of them ("almost settings", or "all settings"). 

.line 129, why you used "centered BMI" ?  (not even standardized).  In general, BMI is used as is, or categoraized, but not centered, in medical research. you need to explain.  

.page 9, MSE on Table 4 is not small, and R2 on Table5 is far from 1. Further, Figure 1 showed 5 data points which cleally not fit to the proposed and CL model.  I felt that your model do not fit well to the data. Line 192, you said that the proposed method provided high prediction accurary compared with CL, but I'm not convinced that the proposed mehtod showed "high" prediction accuracy.  

Reviewer 2 Report

Although the method and the application may be interesting, I find the first part of the article so flawed that I recommend to reject this manuscript just based on this observation.

I am not an expert in multitask learning nor network lasso. Nevertheless, making research in the sparsity area for over 10 years may qualify me to understand what is going on in the presented paper, actually. Yet, the paper is very hard to read for me. I am not saying that the derivations in the framework of ADMM is wrong, I am complaining on the confusing style of writing. Below I make some more detailed comments.

* The abstract is too general.
* English is good at the first sight, but the meaning of some sentences becomes clear only after several rounds of reading. Readability should be significantly improved.
* A few sentences are unclear, for instance "dimensionality...has been increasing". When? Where? Why is this a problem? Is this really the only reason for an l1 regularizer?
* In Sec. 2, the authors say that in general the vectors w_i are assumed to be identical. In my understanding of the problem, such a statement  is in contrast to the optimization problem, where multiple clusters of w are estimated...
* l2,1 regularization is mentioned briefly at the end of Sec. 2. Why? Was this option wrong? What does it mean in comparison to the proposed approach? Isn't it actually more general than the proposed l1 regularization?
* At the end of 4.1, a constrained Weber problem is suddenly introduced. What is the motivation and reason? Why are the authors speaking of "future data" while there was no occurence of this term before?
* There is no comparison to the state of the art methods in the simulation section. Slightly simpler models (CL, SNL) than the proposed one are used in the comparison. It is not argued anywhere that these two are actually the relevant ones.

Minor comments:
* What is a sparse network lasso compared to network lasso? If any, it should be emphasized.
* It is not clear how the graph weights r_{i,j} are constructed/estimated. Yet they play a crucial role in the optimization problem.
* I am missing a link to the publicly available codes and data.
* A sentence should not start with a citation, e.g. [5].
* In Eq. (2), the index i apears in two different meanings.

Round 2

Reviewer 1 Report

Authors had revised the manuscript well, according to reviewer's comments. Therefore, I think this manuscript will be acceptable. 

Although it is not essencial, but I have only one concern as a minor point. Since their method takes the logarithm of the explanatory variables in regression, I was wondering if it wold work well when there is a variable with many zeros. 

Reviewer 2 Report

Dear authors,
thank you for improving your manuscript. I still have a few comments.

Abstract:
by considering heterogeneity come from each sample

Intro:
Mulit  ---> Multi

Section 2:
I think that your answer to my point 5. in your rebuttal should reflect in the manuscript; the reader can have the same doubt as I have. Anyway, it seems to me that your proposed optimization problem (8) is the same as Yamada's problem, except for the power of two, which twists the l1 penalty to the l21 penalty. Notice that your penalty can be seen as a special case of theirs if only one group of w's is considered. In that sense their method is more general than yours. Please explain carefully the relation between the two problems.

Regarding the construction of weights, you should write down that you did it the same way as Yamada (if I haven't overlooked it).

I appreciate that your codes are available publicly. However, concerning the real data, non-public data (not even available to the reviewers) makes a part of your results only to be believed.
